# A Multi-Method Approach to Geophysical Imaging of a Composite Pluton in North Portugal

Cláudia Cruz *, Fernando Noronha and Helena Sant'Ovaia

Departamento de Geociências Ambiente e Ordenamento do Território, Instituto de Ciências da Terra—Pólo Porto, Faculdade de Ciências, Universidade do Porto, Rua do Campo Alegre 687, 4169-007 Porto, Portugal; fmnoronh@fc.up.pt (F.N.); hsantov@fc.up.pt (H.S.)

* Correspondence: claudiacruz@fc.up.pt

**Abstract:** Potassium (K), thorium (Th), and uranium (U) are good markers of magmatic or alteration processes and the surface concentrations of these radioelements can be mapped at the regional or local scale through radiometric (gamma) surveys. In this study, a radiometric survey was performed in a post-orogenic pluton located in North Portugal, namely the Lamas de Olo Pluton, composed by three granitic facies. This pluton has already been intensively studied, including magnetic susceptibility, gravimetric, geochemical, and petrographic studies. The main objective of this work is to evaluate the radiometric data and combine them with other characteristics, such as magnetic susceptibility, and gravimetry, as well as to elucidate structures such as faults and fractures, outline geological boundaries, and identify alteration zones within various granites of the pluton. The radiometric approach reveals the spatial distribution of radioelements, offering a more distinct portrayal of the geology in the studied area. The radioactive heat production rate was calculated for the studied pluton, showing that the mean value is 4.09 µW m$^{-3}$, surpassing the known mean values for granites. Our study highlights that radiometric measurements unveil compositional variations within granitic pluton and aid in identifying feeder zones. Furthermore, these measurements can be correlated with each type of granites, demonstrating associations with surface concentrations of K-Th-U. Our findings indicate a spatial alignment between the NE feeder root and a U-rich granite (Barragem granite), as evidenced by its elevated concentration of this radioelement. Conversely, the other root displays a notable relative concentration of Th, consistent with the Th-rich characteristics observed of the two other granites (Lamas de Olo and Alto dos Cabeços granites).

**Keywords:** Variscan granites; radiometric data; radioelements; gravimetric survey





## 1. Introduction

Geophysical techniques form an integral part of most geological mapping works and have a wide application in reconnaissance surveys. In recent years, there has been an increasing use of geophysical methods, owing to their ability to map and delineate subtle physical variations in the geology most often missed during fieldwork (e.g., [1–6]).

Geophysical methods used in this work include radiometry, magnetic susceptibility, and gravimetry. These methods, like other geophysical techniques, use specific physical properties such as radioactivity, magnetic susceptibility, and density to map the characteristics of the outcropping and non-outcropping rocks. Both magnetic susceptibility and gravity surveys are useful for delineating geological boundaries. However, gravimetric surveys have a greater depth of investigation and are commonly used to map geological features at a deeper level, allowing us to infer the geometry of a pluton. Concerning radiometric surveys, they are useful in geological mapping, since the different types of rock can be recognized from their distinctive radioactive signature [7,8].

Radiometric surveys measure variations in three radioelements, namely potassium (K%), thorium (Th ppm), and uranium (U ppm) in the soil and exposed rocks. It is important

to understand that the radiometric method has several characteristics that make it unique amongst the geophysical methods, namely: (i) the measured radioactivity emanates only from the Earth's surface's top few centimeters, and (ii) the identification of elemental sources through the energy of emitted $\gamma$-rays allows radiometric data to map variations in the chemical, rather than physical, characteristics of the studied area [2]. The total gamma radiation constitutes a "frozen image" of the last stage of the magmatic building of plutons and is statistically reliable for distinguishing between rock units (e.g., [1]). Therefore, variations in radiation concentration are related to the presence of lithologies with different chemical and mineralogical composition.

During radioactive decay, mass is converted to energy [9]. The radioactive decay of K, Th, and U is recognized as an important heat source in continental crust with typical heat-production values of 0.1–3.0 $\mu$W m$^{-3}$ (e.g., [10,11]).

The generation of heat within a particular rock is influenced by the concentrations of K, Th, and U, which exhibit significant variations depending on the rock type. However, certain patterns emerge due to the consistent geochemical behavior of these elements during processes such as magmatic differentiation, sedimentation, and metamorphism, which contribute to the distribution of these radioelements in rocks. In the case of granites, the heat production varies between 1.25 $\pm$ 0.83 $\mu$W m$^{-3}$ and 4.36 $\pm$ 2.17 $\mu$W m$^{-3}$ [12], and the average value is reported to be ca. 2.00 $\mu$W m$^{-3}$ in [12] studies, and 2.45 $\mu$W m$^{-3}$ according to [13].

The radioactive heat production rate in a rock can be calculated using the following equation [9]:

$$H = 10^{-5}\rho \left(9.52c_U + 2.56c_{Th} + 3.48\,c_K,\ \text{in } \mu\text{W m}^{-3} \right) \tag{1}$$

where $H$ represents the radioactive heat production rate, $\rho$ is density (in kg m$^{-3}$), $9.52 \times 10^{-5}$, $2.56 \times 10^{-5}$ and $3.48 \times 10^{-9}$ are the heat production constants for U, Th, and K, respectively (in W kg$^{-1}$), and the $c_U$, $c_{Th}$, and $c_K$ are the measured content of U (ppm), Th (ppm), and K (%).

In the present work, the geophysical methods were applied to post-orogenic Variscan granites from the Lamas de Olo pluton. The magnetic susceptibility and gravimetric studies were previously performed by Cruz et al. [5,14,15], and the radiometric approach is now overlapped and analyzed together with the data previously obtained.

The main objective of this work is to map the radiometric response of the Lamas de Olo pluton, emphasizing the correlation between new radiometric data and the magnetic characteristics of the pluton's facies, its geometry in depth, and the positioning of its feeding zones. The overarching goal is to enhance our understanding of this pluton.

## 2. Geological Setting

The Iberian Variscan belt, formed by the collision of Laurussia and Gondwana during the Devonian–Carboniferous times, is a significant curved section within the broader European Variscan belt [16–18]. In Portugal, three main ductile deformation phases (D$_1$, D$_2$, and D$_3$) have been identified [19]. In the Central Iberian Zone (CIZ), only D$_1$ (ca. 360–337 Ma; e.g., [20–24]), and D$_3$ (ca. 321–300 Ma; e.g., [24,25]) were distinguished. D$_1$ resulted in isoclinal folds with a sub-horizontal axial plane, NW-SE strike, and D$_3$ characterized by wide folding with small amplitude, featuring a subvertical axial plane and NW-SE sub-horizontal axis. Simultaneously, during D$_3$, vertical ductile shear zones developed in lower structural levels, while upper levels experienced brittle deformation, leading to fracture-conjugate systems with NNE-SSW and NNW-SSE orientations [26]. The late-Variscan period is recognized as a time of widespread fracturing in the Variscan belt of southwest Europe [26–28].

Northwestern Portugal is occupied by a large volume of granites emplaced during the post-collisional stage of the Variscan orogeny (312–290 Ma; e.g., [24,25,29]).

The target of this study is a pluton located in the northern part of the Central Iberian Zone, the Lamas de Olo pluton, in northern Portugal (LOP, Figure 1).

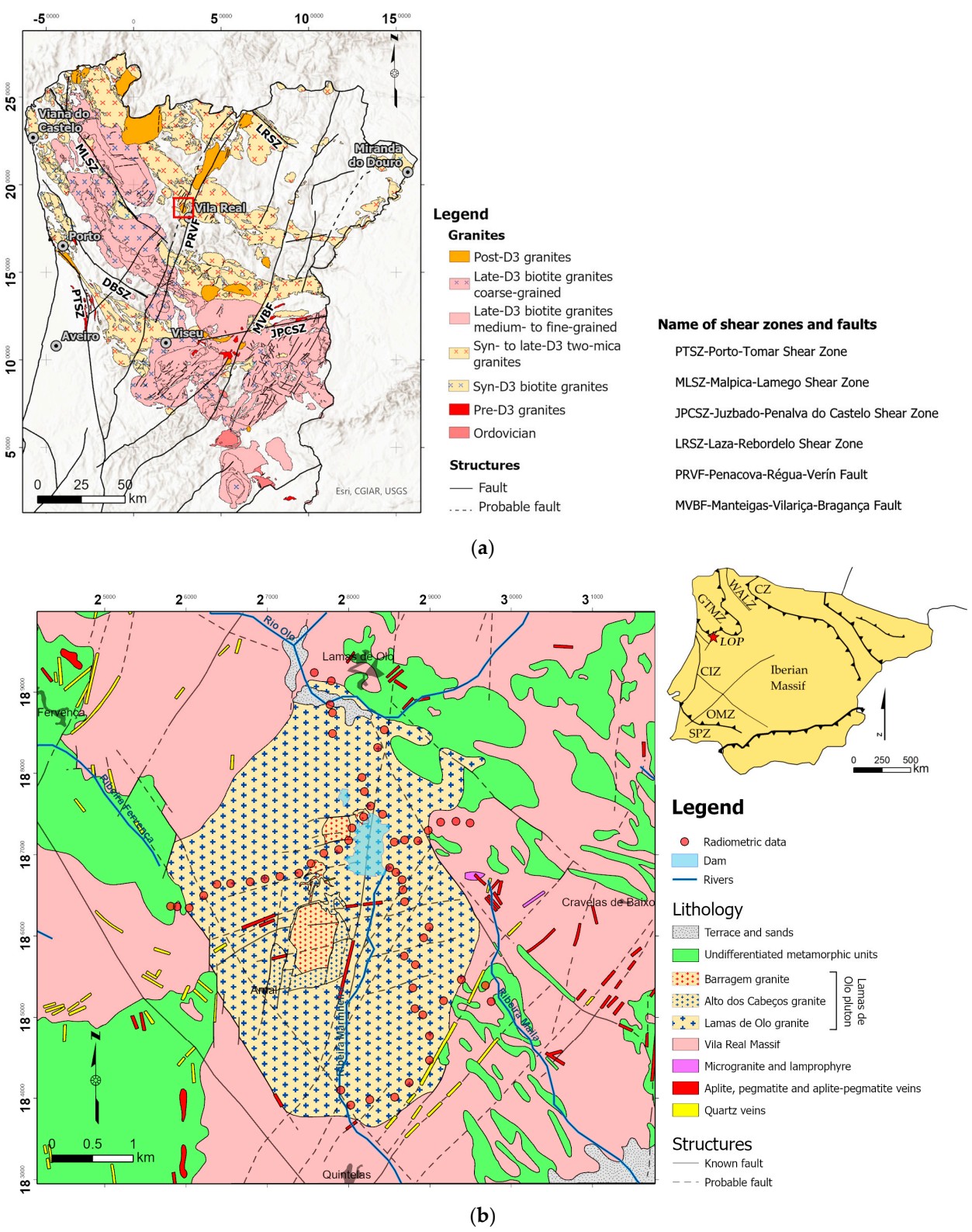

**Figure 1.** (**a**) Simplified geologic map of northern and central Portugal with the plutons and principal faults and shear zones (adapted from Ferreira et al. [25]); the red square indicates the pluton under study: the Lamas de Olo pluton. (**b**) Simplified geological map of the Lamas de Olo Pluton (modified: [30,31]) with radiometric measurement points; in the Iberian Variscan belt the location of the studied pluton was identified by a red star. LOP—Lamas de Olo Pluton. CZ—Cantabric Zone, WALZ—West Asturian Leonese Zone, GTMZ—Galicia Trás-os-Montes Zone, CIZ—Central Iberian Zone, OMZ—Ossa Morena Zone, SPZ—Sul Portuguesa Zone (based on: [32,33]).

The LOP is a composite pluton constituted by three outcropping biotite granites, similar in mineralogical composition but with different textures and different biotite contents: (i) Lamas de Olo (LO), (ii) Alto dos Cabeços (AC), and (iii) Barragem (BA) granites. LOP granites are mostly composed of quartz, plagioclase (oligoclase, oligoclase–andesine, and albite), K-feldspars (perthitic orthoclase, and microcline), and biotite. Muscovite I (Ms I), muscovite II (Ms II; post-magmatic muscovite), zircon, sphene, allanite, fluorite, hematite, magnetite, ilmenite, chlorite, rutile, apatite, goethite, epidote, and tourmaline are present as accessory minerals (e.g., [14]). The LO granite is the most abundant in outcrop and is characterized by a medium-to-coarse-grained porphyritic granite (Biotite (Bt) > Muscovite (Ms)). The AC corresponds to a fine-to-medium-grained porphyritic granite (Bt > Ms). The younger granite, which cut the other two granites, is BA which outcrops in the center of the pluton and near a dam and is classified as leucocratic fine-to-medium-grained and slightly porphyritic granite (Bt = Ms I + Ms II) [14,34,35].

The LOP intrudes metasediments from the Douro Group (DG) of the Schist–Graywacke Complex supergroup (CXG), and syntectonic (syn-$D_3$) peraluminous granites from Vila Real Massif (VRM) (Figure 1; [34]). The VRM is a composite massif constituted by two-mica granites, with the monazite and zircon analysis yielding a weighted average $^{207}Pb/^{235}U$ age of ca. $311 \pm 1$ Ma [36]. In the field, the contacts between LO and AC granites are generally diffuse; however, field observations show that BA granite crosscuts the LO and the AC granites. $^{207}Pb/^{235}U$ dating yields an age of $297.19 \pm 0.73$ Ma to LO granite [35].

Previous studies (e.g., [5]) show that this pluton has a rhombus shape controlled by an NNW-SSE fault system conjugated with both the NNE-SSW and WNW-ESE system faults.

The NNE-SSW system is parallel to the Penacova-Régua-Verin fault, still active and marked by hot $CO_2$-rich mineral water springs [37,38]. The gas-carbonic springs are reported in the localities of Chaves, Vidago, and Pedras Salgadas [37,38]. However, there is no evidence of water springs in the studied area. Regarding the WNW-ESE system fault, commonly affected by late episyenitization processes, this occurs mostly in the ESE border of the pluton. Beyond that, W-Mo(-Sn) mineralizations occurred in association with some of the WNW-ESE quartz veins [5,31].

Recently, and based on the new classification of Hildenbrand et al. [39], LOP is classified as a late-$C_3$ pluton. Expanding on this, Sant'Ovaia et al. [40] delve into the magnetic fabric and show that LOP is an example of a pluton emplaced during the final stage of collision, controlled by regional late $C_3$, NNW-SSE strike-slip faults within a wrench regime.

### 3. Materials and Methods

*3.1. Radiometric Imaging*

Two types of data were analyzed in this work: (i) available radiometric data, and (ii) data collected in the field with specific equipment.

(i) Available radiometric data

The radiometric data were provided by "Unidade de Recursos Minerais e Geofísica" from "Laboratório de Geologia e Minas—Laboratório Nacional de Energia e Geologia" (LGM-LNEG; [41]). The data correspond to an area that covers the "Folha 101—Lordelo-Vila Real" from the Military Map of Portugal at a scale of 1:25,000. LNEG provided 4118 radiometric measurements of total gamma radiation (nGy h$^{-1}$).

Different methodologies were used to obtain these radiometric measurements, namely using a scintillator SAPHYMO SRAT, model SPP2. Complementary surveys were made in regional scaling, using spectrometers such as Exploranium GR-256 hand held, and Exploranium GR-650 4 L volume of crystal coupled to a car [41].

(ii) Radiometric survey in the field

The instrument employed for radiometric surveying is referred to as a gamma-ray spectrometer, and in this study, the portable GR-320 Exploranium model (Manufactured by SAIC Exploranium, 6108 Edwards Blvd, Mississauga, ON, Canada) was used. The portable

equipment provides measurements for individual isotopes of K (in %), Th (in ppm), and U (in ppm).

The radiometric survey comprised 57 measuring stations (Figure 1), with three measurements conducted at each station and subsequent calculation of the averages. In the area studied, 48 measurements are performed in LOP granites, and only 9 measurements corresponded to the lithologies of the LOP surrounding rocks (7 in VRM, and 2 in CXG).

The terrain in the study area is characterized by hills, making it challenging to reach numerous outcrops. Accordingly, data collection primarily focused on considering local roads and access points along two main directions, N-S and W-E. Therefore, while the analysis includes the entire pluton, it should be noted that the NW and SW areas of the pluton, as well as the central area of the pluton, composed by BA and AC granite, pose greater challenges in terms of data assessment. Consequently, any conclusions drawn should not be considered definitive but rather justify detailed examination.

### 3.2. Magnetic Susceptibility and Gravimetric Approach

These techniques were previously explained, and their results were discussed in Cruz et al. [5,14,15].

The magnetic susceptibility measurements were performed using the KLY-4S Kappabridge susceptometer Agico model (Manufactured by Advanced Geoscience Instruments Company, Agico, situated in Brno, Czech Republic) from the Instituto de Ciências da Terra—Polo Porto. At the lab, the data were gathered in a reversible magnetization field, at 0.3 mT and room temperature. The magnetic susceptibility was measured in 350 oriented cores from 48 sampling sites roughly distributed across Lamas de Olo granites [5,14,15].

The gravity measurements were performed using a LaCoste & Romberg G1054 gravimeter (Manufacturered by LaCoste & Romberg, Austin, TX, USA) from the Instituto de Ciências da Terra—Polo do Porto. In total, 271 gravimetric stations were measured along 3 gravimetric profiles both on the LOP granites and host rocks [5]. However, the comparison between magnetic susceptibility, gravimetric response, and radiometric imaging has not been performed yet.

### 3.3. Coordinate System and Map Specification

The data are in the PT-TM06/ETRS89 coordinate system and the map interpolation was made using Oasis Montaj 2023.2 and ArcGIS PRO 3.2.2 software. The minimum curvature interpolation method was used to create all the interpolated maps in this work, namely: magnetic susceptibility and residual anomaly maps, as well as all the maps related to the radiometric data. The white lines present in Figures 2–7 represent the isolines of the residual anomaly (3rd order) and are superimposed to help the interpretation of the radiometric data.

## 4. Results and Discussion

### 4.1. Radiometric Data Interpretation

As a starting approach, the gamma radiation map was interpolated using data provided by LNEG (Figure 2).

The radiometric map (Figure 2) shows the higher values of gamma radiation in the SE zone of the pluton.

Since the provided data only present the total radiation without the individual spectra of K, Th, and U, a more detailed study of the LOP granites has been carried out.

A radiometric survey was conducted, mostly in the LO granite, and the data were analyzed. Several interpolation maps were created using the obtained data: (i) a map displaying the distribution of total radiation values (Figure 3a); (ii) a map showing the distribution of K isotope values (%; Figure 3b); (iii) a map corresponding to the interpolation of the Th isotope values (ppm; Figure 3c); (iv) a map depicting the distribution of U isotope values (ppm; Figure 3d), and (v) a red–green–blue (RGB) ternary map (Figure 4).

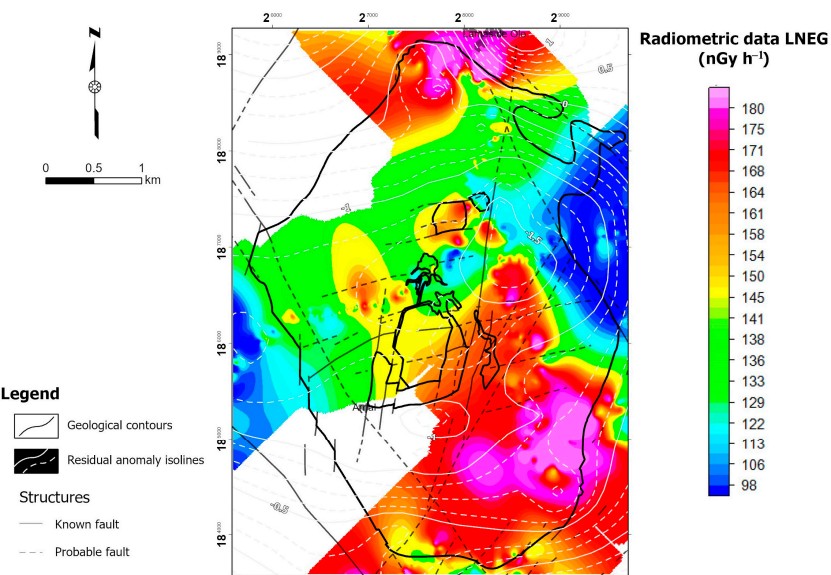

**Figure 2.** Interpolation of radiometric data in nGy h$^{-1}$ (values provided by LNEG) in the LOP (Radiometric data interpolation made by minimum curvature in the Oasis Montaj 2023.2 software: modified from: [41]).

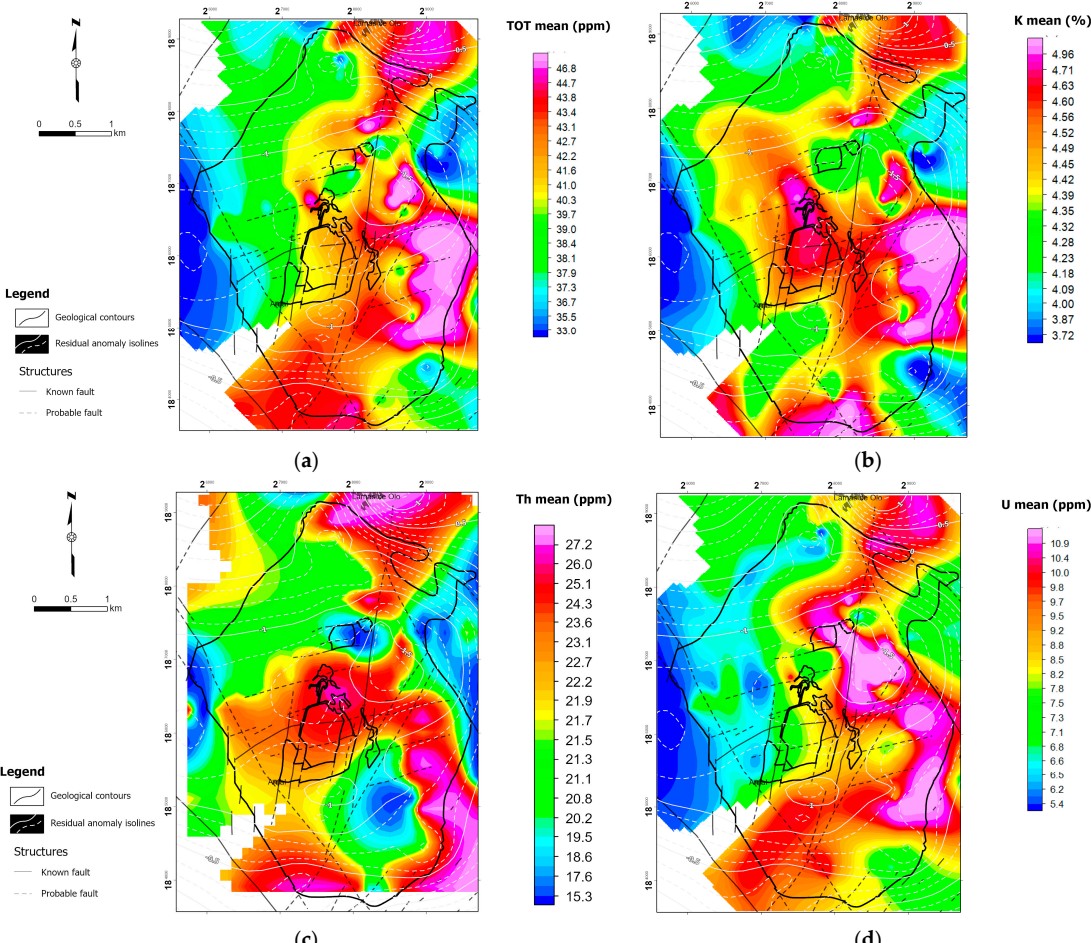

**Figure 3.** Radiometric data interpolated maps of the distribution of (**a**) total radiation values (ppm); (**b**) K values (%); (**c**) Th values (ppm); and (**d**) U values (ppm; radiometric data interpolation made by minimum curvature in the Oasis Montaj 2023.2 software).

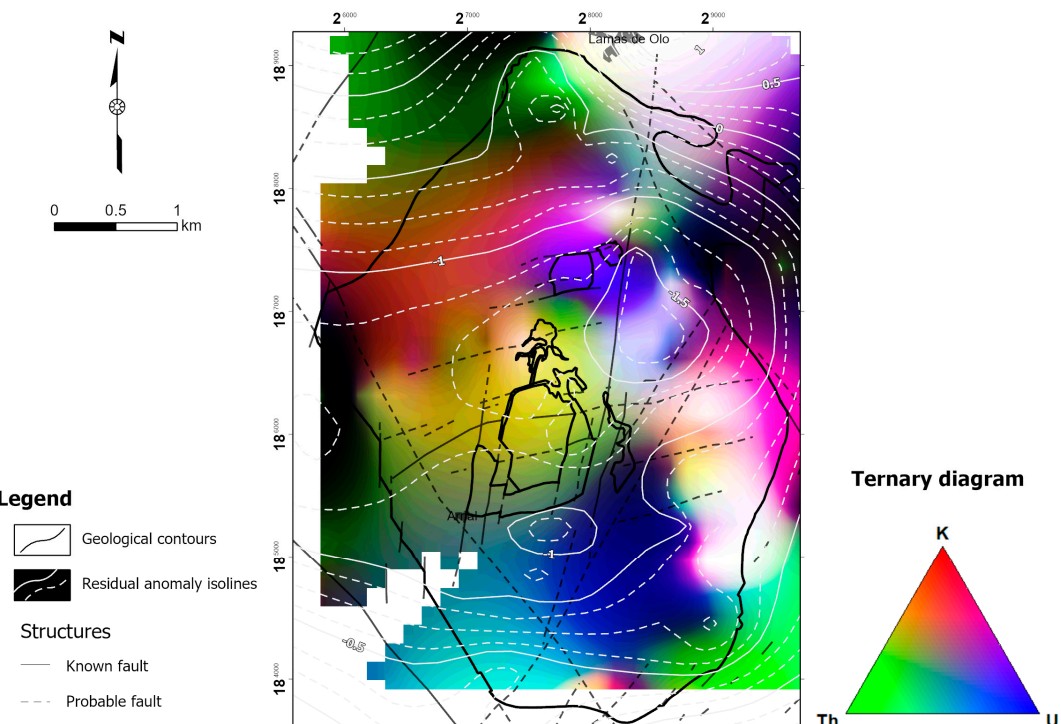

**Figure 4.** Composite ternary RGB radiometric map, with K, Th, and U in the red (R), green (G), and blue (B) channels, respectively (map made in the Oasis Montaj 2023.2 software).

As an alternative to the four concentration maps (Figure 3), Figure 4 shows a commonly used approach to visualize gamma-ray spectrometry data, allowing the simultaneous representation of three isotopes. The ternary K–Th–U radiometric map was created through a red–green–blue (RGB) additive chromatic combination applied point by point for the three radiometric measurements. The ternary map was created through the "Grid and Image" tool of the Oasis Montaj 2023.2 software. Despite the impact of weathering and soil development on the area, the ternary K–Th–U radiometric maps can effectively display geological features [1].

In the following discussion concerning the ternary K–Th–U radiometric map, terms such as "K-relative rich" or "K-relative poor" do not depict absolute variations in K content but rather indicate high or low K/(K + Th + U) ratios. This principle similarly applies to the analysis of Th and U content. In the NE section of the ternary map, a blueish zone is evident, reflecting a U-relative high concentration compared to K and Th isotopes. This zone corresponds to the northern outcrop of Barragem granite. Another light blue zone at the south is identified, corresponding to a relatively high U and Th concentration, and depleted in K. At the north, a greenish zone appears, corresponding to a Th-relative high concentration. The reddish zone located at NW in the map suggests the presence of an area richer in K. However, this area is difficult to assess, and no data have been collected on site. Accordingly, interpretations rely on the interpolation of available data, emphasizing the need to refrain from drawing direct conclusions. Much of the eastern part of the ternary map comprises white tones, which are characteristic of a relative high radioelement concentration and represent an area intensely fractured in the Lamas de Olo granite. Yellowish tones appear in the center of LOP, which means that this zone has a relatively high K-Th concentration. It is important to note that this region is predominantly occupied by the BA and AC granites. However, no specific measurements were obtained on these granites. The map is generated through data interpolation based on measurements from the LO granite. Therefore, the observed enrichment in K-Th and relative depletion in U are characteristic features of the LO granite that outcrops in this area. Otherwise, in the

western area of LOP, the green dark color is dominant, which corresponds to a relatively low radioelement concentration (Figure 4).

Table 1 displays the descriptive statistics for values obtained across the study area, as well as specifically for the LOP granites and surrounding rocks (mostly VRM). The total radiation data (the sum of K-Th-U isotopes concentration) range between 27.7 and 59.9 ppm (Table 1). Comparing the interpolation maps of the total gamma radiation data (Figure 3a) with the map of total gamma radiation interpolated with the LNEG data, some similarities are noticeable (Figure 2).

**Table 1.** Descriptive statistics of radiometric data in the study area ($\overline{x}$: mean value; $\sigma$: standard deviation; min.: minimum value; max.: maximum value).

| | | Total (ppm) | K (%) | Th (ppm) | U (ppm) | eU/eTh | eTh/K | eU/K |
|---|---|---|---|---|---|---|---|---|
| LOP + surrounding rocks | $\overline{x}$ | 41.1 | 4.4 | 22.1 | 8.8 | 0.4 | 5.0 | 2.0 |
| | $\sigma$ | 6.2 | 0.6 | 3.9 | 2.7 | 0.1 | 0.5 | 0.5 |
| | Min. | 27.7 | 2.8 | 12.9 | 3.3 | 0.2 | 3.3 | 0.9 |
| | Max. | 59.9 | 6.0 | 30.3 | 18.5 | 1.2 | 6.6 | 3.9 |
| LOP | $\overline{x}$ | 42.0 | 4.5 | 22.5 | 9.1 | 0.4 | 5.0 | 2.0 |
| | $\sigma$ | 6.0 | 0.6 | 3.7 | 2.7 | 0.1 | 0.6 | 0.5 |
| | Min. | 27.7 | 2.8 | 13.1 | 5.1 | 0.2 | 3.3 | 1.3 |
| | Max. | 59.9 | 6.0 | 30.3 | 18.5 | 1.2 | 6.6 | 3.9 |
| Surrounding rocks | $\overline{x}$ | 36.1 | 4.0 | 19.7 | 6.8 | 0.4 | 4.9 | 1.7 |
| | $\sigma$ | 5.0 | 0.5 | 4.1 | 1.6 | 0.1 | 0.8 | 0.3 |
| | Min. | 27.8 | 3.2 | 12.9 | 3.3 | 0.2 | 3.9 | 0.9 |
| | Max. | 43.8 | 4.7 | 24.6 | 8.9 | 0.5 | 6.0 | 1.9 |

Looking to the values of LOP granites, for the K isotope, the minimum value is 2.8% and the maximum is 6.0%; for Th, the minimum is 13.1 ppm and the maximum is 30.3 ppm; for U, the minimum is 5.1 ppm and the maximum is 18.5 ppm. The Th is the isotope that has the higher variation in the LOP granites (the standard deviation is 3.7 ppm), and K is the one that has the lower variability (the standard deviation is 0.6%; Table 1).

A detailed examination of the radiometric data interpolation maps and the geological map reveals that:

- Both total radiation maps, the one using LNEG data interpolation (Figure 2) and the other using data obtained with portable device (Figure 3a), reveal significant variability across the pluton. The high concentration is located at the east border of LOPl;
- The higher concentration of Th is spatially related to areas with intense fracturing, coinciding with locations where NNE-SSW and NE-SW fractures intersect with WNW-ESE structures. This correlation is particularly notable in the middle of pluton, and at NE and ESE borders of LOP (Figures 1 and 3c);
- The U concentration is higher in the eastern area of LOP, exhibiting a NNE-SSW trend. A closer examination shows that the U concentration is highest in the ESE zone of the pluton, where there is intense WNW-ESE fracturing. Additionally, the high U concentration is spatially related to the NNE-SSW fault, parallel to the regional Penacova-Régua-Verin fault (Figures 1 and 3d).

While radiometric data offer insights into isotopic concentration and whole-rock geochemical analysis provides information on the chemical and mineralogical composition of rocks, it is possible to establish a connection between the two.

Previous geochemical studies [15] reveal that LOP granites exhibit characteristics of being calc-alkaline to alkali-calcic as well as peraluminous in nature. Regarding to the rare earth element (REE) spectra, all granites display a flat pattern, with a slight pronounced Eu anomaly in LO and AC granites and a more pronounced Eu anomaly in BA granite. The findings imply the presence of two distinct magma sources: LO and AC granites have

geochemical similarities, suggesting a common magma origin, while BA granite originates from a different magma source and represents the most evolved granite. The BA granite, characterized as an alkaline granite, is the youngest and most evolved, displaying REE spectra with a similar pattern to anorogenic granites. These observations suggest that BA granite was the last to ascend and be emplaced in the crust.

The geochemical data (Table 2) show that the major element of $K_2O$ (%) has similar values in all the LOP granites ($\sigma = 0.24$) which is in accordance with the variability of K isotope (Figure 3b and Table 1). Regarding the Th average content, is quite similar in the Lamas de Olo ($\overline{x} = 15.2$ ppm) and Alto dos Cabeços ($\overline{x} = 16.5$ ppm) granites, having the lower values in the Barragem ($\overline{x} = 9.4$ ppm) granite. Otherwise, the U element mean values are much higher in the Barragem ($\overline{x} = 28.0$ ppm) granite, reaching values of 38.9 ppm, while in other granites the values are lower (LO: $\overline{x} = 10.0$ ppm; AC: $\overline{x} = 7.4$ ppm) (Table 2; [10]).

**Table 2.** Descriptive statistics of major and trace elements in LOP granites based on geochemical data ($\overline{x}$: mean value; $\sigma$: standard deviation; Min.: minimum value; Max.: maximum value; adapted from: Cruz et al. [15]).

| | Lamas de Olo Granite | | | | Alto dos Cabeços Granite | | | | Barragem Granite | | | | LOP | | | |
|---|---|---|---|---|---|---|---|---|---|---|---|---|---|---|---|---|
| | $\overline{x}$ | $\sigma$ | Min. | Max. | $\overline{x}$ | $\sigma$ | Min. | Max. | $\overline{x}$ | $\sigma$ | Min. | Max. | $\overline{x}$ | $\sigma$ | Min. | Max. |
| $SiO_2$ (%) | 73.63 | 1.41 | 71.69 | 75.04 | 72.50 | 1.27 | 71.11 | 73.60 | 76.84 | 0.73 | 76.23 | 77.65 | 74.15 | 2.04 | 71.11 | 77.65 |
| $TiO_2$ (%) | 0.185 | 0.02 | 0.168 | 0.206 | 0.241 | 0.03 | 0.215 | 0.279 | 0.091 | 0.007 | 0.084 | 0.098 | 0.175 | 0.059 | 0.084 | 0.279 |
| $Al_2O_3$ (%) | 13.72 | 0.29 | 13.30 | 14.04 | 13.50 | 0.33 | 13.21 | 13.86 | 12.34 | 0.03 | 12.31 | 12.37 | 13.32 | 0.64 | 12.31 | 14.04 |
| $Fe_2O_3$ (%) | 0.95 | 0.18 | 0.60 | 1.13 | 0.77 | 0.18 | 0.64 | 0.97 | 0.30 | 0.10 | 0.21 | 0.41 | 0.74 | 0.32 | 0.21 | 1.13 |
| FeO (%) | 1.0 | 0.2 | 0.7 | 1.2 | 1.2 | 0.1 | 1.1 | 1.3 | 0.7 | 0.2 | 0.5 | 0.9 | 0.9 | 0.2 | 0.5 | 1.3 |
| $Fe_2O_3$ (T%) | 2.01 | 0.12 | 1.91 | 2.23 | 2.07 | 0.31 | 1.87 | 2.42 | 1.08 | 0.30 | 0.83 | 1.41 | 1.79 | 0.48 | 0.83 | 2.42 |
| MnO (%) | 0.059 | 0.01 | 0.039 | 0.067 | 0.060 | 0.02 | 0.043 | 0.075 | 0.031 | 0.018 | 0.020 | 0.052 | 0.052 | 0.018 | 0.020 | 0.075 |
| MgO (%) | 0.33 | 0.05 | 0.29 | 0.41 | 0.50 | 0.08 | 0.46 | 0.59 | 0.16 | 0.01 | 0.16 | 0.17 | 0.33 | 0.13 | 0.16 | 0.59 |
| CaO (%) | 0.98 | 0.14 | 0.73 | 1.16 | 1.26 | 0.26 | 0.96 | 1.42 | 0.39 | 0.14 | 0.29 | 0.55 | 0.90 | 0.37 | 0.29 | 1.42 |
| $Na_2O$ (%) | 3.49 | 0.13 | 3.39 | 3.74 | 3.00 | 0.38 | 2.56 | 3.23 | 3.10 | 0.09 | 2.99 | 3.15 | 3.27 | 0.30 | 2.56 | 3.74 |
| K2O (%) | 4.69 | 0.21 | 4.39 | 4.98 | 4.26 | 0.09 | 4.18 | 4.36 | 4.52 | 0.14 | 4.41 | 4.67 | 4.54 | 0.24 | 4.18 | 4.98 |
| $P_2O_5$ (%) | 0.06 | 0.01 | 0.05 | 0.07 | 0.07 | 0.01 | 0.07 | 0.08 | 0.04 | 0.01 | 0.03 | 0.04 | 0.06 | 0.02 | 0.03 | 0.08 |
| LOI (%) | 0.54 | 0.15 | 0.39 | 0.73 | 1.28 | 0.81 | 0.66 | 2.19 | 0.81 | 0.26 | 0.51 | 0.99 | 0.79 | 0.49 | 0.39 | 2.19 |
| Total (%) | 99.58 | 0.85 | 98.34 | 100.5 | 98.63 | 0.35 | 98.31 | 99.00 | 99.32 | 0.57 | 98.94 | 99.97 | 99.28 | 0.76 | 98.31 | 100.5 |
| Sc (ppm) | 5 | 0 | 5 | 5 | 5 | 0 | 5 | 5 | 5 | 1 | 4 | 5 | 5 | 0 | 4 | 5 |
| Be (ppm) | 5 | 2 | 4 | 10 | 7 | 3 | 5 | 11 | 48 | 73 | 6 | 132 | 17 | 36 | 4 | 132 |
| V (ppm) | 15 | 1 | 14 | 17 | 22 | 3 | 20 | 25 | 8 | 2 | 7 | 10 | 15 | 5 | 7 | 25 |
| Cr (ppm) | 117 | 28 | 80 | 150 | 110 | 17 | 100 | 130 | 127 | 117 | 40 | 260 | 118 | 54 | 40 | 260 |
| Co (ppm) | 3 | 0 | 3 | 3 | 3 | 0 | 3 | 3 | 1 | 1 | 1 | 2 | 3 | 1 | 1 | 3 |
| Ni (ppm) | 13 | 8 | 10 | 30 | 33 | 6 | 30 | 40 | 37 | 38 | 10 | 80 | 24 | 21 | 10 | 80 |
| Cu (ppm) | 5 | 0 | 5 | 5 | 20 | 10 | 10 | 30 | 60 | 69 | 20 | 140 | 23 | 38 | 5 | 140 |
| Zn (ppm) | 34 | 10 | 15 | 40 | 47 | 6 | 40 | 50 | 55 | 49 | 15 | 110 | 43 | 24 | 15 | 110 |
| Ga (ppm) | 18 | 1 | 17 | 18 | 18 | 1 | 17 | 18 | 17 | 1 | 17 | 18 | 18 | 1 | 17 | 18 |
| Ge (ppm) | 2 | 0 | 2 | 3 | 2 | 0 | 2 | 2 | 3 | 1 | 2 | 3 | 2 | 0 | 2 | 3 |
| As (ppm) | 3 | 0 | 3 | 3 | 3 | 0 | 3 | 3 | 5 | 2 | 3 | 7 | 3 | 2 | 3 | 7 |
| Rb (ppm) | 242 | 15 | 223 | 265 | 253 | 29 | 221 | 277 | 304 | 15 | 291 | 321 | 260 | 32 | 221 | 321 |
| Sr (ppm) | 98 | 13 | 81 | 114 | 147 | 5 | 142 | 152 | 27 | 9 | 21 | 37 | 93 | 46 | 21 | 152 |
| Y (ppm) | 33 | 3 | 29 | 37 | 25 | 7 | 19 | 33 | 37 | 13 | 28 | 52 | 32 | 8 | 19 | 52 |
| Zr (ppm) | 107 | 7 | 100 | 118 | 123 | 14 | 113 | 139 | 50 | 7 | 45 | 58 | 97 | 30 | 45 | 139 |
| Nb (ppm) | 18 | 1 | 16 | 20 | 18 | 2 | 16 | 20 | 37 | 11 | 27 | 48 | 23 | 10 | 16 | 48 |
| Mo (ppm) | 3 | 1 | 2 | 4 | 2 | 0 | 2 | 2 | 4 | 2 | 2 | 6 | 3 | 1 | 2 | 6 |
| Ag (ppm) | 0.3 | 0.1 | 0.3 | 0.5 | 0.3 | 0.0 | 0.3 | 0.3 | 0.4 | 0.2 | 0.3 | 0.6 | 0.3 | 0.1 | 0.3 | 0.6 |
| In (ppm) | 0.1 | 0.0 | 0.1 | 0.1 | 0.1 | 0.0 | 0.1 | 0.1 | 0.1 | 0.0 | 0.1 | 0.1 | 0.1 | 0.0 | 0.1 | 0.1 |
| Sn (ppm) | 7 | 2 | 3 | 10 | 9 | 4 | 6 | 13 | 9 | 4 | 7 | 14 | 8 | 3 | 3 | 14 |
| Sb (ppm) | 0.3 | 0.0 | 0.3 | 0.3 | 0.3 | 0.0 | 0.3 | 0.3 | 0.3 | 0.0 | 0.3 | 0.3 | 0.3 | 0.0 | 0.3 | 0.3 |

**Table 2.** *Cont.*

| | Lamas de Olo Granite | | | | Alto dos Cabeços Granite | | | | Barragem Granite | | | | LOP | | | |
|---|---|---|---|---|---|---|---|---|---|---|---|---|---|---|---|---|
| | $\bar{x}$ | $\sigma$ | Min. | Max. | $\bar{x}$ | $\sigma$ | Min. | Max. | $\bar{x}$ | $\sigma$ | Min. | Max. | $\bar{x}$ | $\sigma$ | Min. | Max. |
| Cs (ppm) | 11.1 | 3.8 | 7.6 | 18.5 | 13.9 | 4.1 | 10.3 | 18.4 | 11.8 | 2.3 | 9.4 | 13.9 | 12.0 | 3.5 | 7.6 | 18.5 |
| Ba (ppm) | 388 | 78 | 282 | 483 | 531 | 53 | 481 | 587 | 81 | 45 | 50 | 132 | 347 | 182 | 50 | 587 |
| Hf (ppm) | 3.3 | 0.3 | 3.0 | 3.8 | 3.7 | 0.2 | 3.5 | 3.9 | 2.6 | 0.1 | 2.5 | 2.7 | 3.2 | 0.5 | 2.5 | 3.9 |
| Ta (ppm) | 4.3 | 0.6 | 3.7 | 5.2 | 3.7 | 1.2 | 2.8 | 5.0 | 10.9 | 2.9 | 7.8 | 13.6 | 5.8 | 3.4 | 2.8 | 13.6 |
| W (ppm) | 4 | 2 | 2 | 8 | 3 | 0 | 3 | 3 | 7 | 3 | 4 | 9 | 5 | 3 | 2 | 9 |
| Tl (ppm) | 1.3 | 0.1 | 1.2 | 1.4 | 1.4 | 0.1 | 1.3 | 1.5 | 1.5 | 0.2 | 1.3 | 1.6 | 1.4 | 0.1 | 1.2 | 1.6 |
| Pb (ppm) | 31 | 1 | 29 | 32 | 29 | 1 | 28 | 30 | 41 | 6 | 35 | 47 | 33 | 6 | 28 | 47 |
| Bi (ppm) | 0.4 | 0.3 | 0.2 | 0.8 | 0.2 | 0.0 | 0.2 | 0.2 | 6.0 | 8.6 | 0.7 | 16.0 | 1.7 | 4.5 | 0.2 | 16.0 |
| Th (ppm) | 15.2 | 1.8 | 13.2 | 18.3 | 16.5 | 1.2 | 15.6 | 17.9 | 9.4 | 0.1 | 9.4 | 9.5 | 14.1 | 3.1 | 9.4 | 18.3 |
| U (ppm) | 10.0 | 5.6 | 5.6 | 20.9 | 7.4 | 1.8 | 5.4 | 9.0 | 28.0 | 14.1 | 12.1 | 38.9 | 13.9 | 11.2 | 5.4 | 38.9 |
| La (ppm) | 20.6 | 2.9 | 18.5 | 26.0 | 29.5 | 3.5 | 27.2 | 33.5 | 12.1 | 0.2 | 12.0 | 12.3 | 20.7 | 6.9 | 12.0 | 33.5 |
| Ce (ppm) | 42.3 | 5.1 | 36.4 | 50.6 | 55.4 | 5.1 | 50.2 | 60.3 | 22.9 | 5.8 | 17.4 | 28.9 | 40.7 | 13.0 | 17.4 | 60.3 |
| Pr (ppm) | 4.82 | 0.57 | 4.28 | 5.86 | 6.51 | 0.52 | 6.19 | 7.11 | 3.12 | 0.37 | 2.82 | 3.53 | 4.82 | 1.34 | 2.82 | 7.11 |
| Nd (ppm) | 18.5 | 2.3 | 16.6 | 22.8 | 23.4 | 1.1 | 22.6 | 24.7 | 11.5 | 1.2 | 10.6 | 12.9 | 18.0 | 4.8 | 10.6 | 24.7 |
| Sm (ppm) | 4.5 | 0.6 | 3.9 | 5.6 | 4.8 | 0.3 | 4.6 | 5.1 | 3.6 | 0.5 | 3.0 | 3.9 | 4.3 | 0.7 | 3.0 | 5.6 |
| Eu (ppm) | 0.58 | 0.04 | 0.54 | 0.64 | 0.71 | 0.04 | 0.68 | 0.75 | 0.22 | 0.04 | 0.18 | 0.25 | 0.52 | 0.19 | 0.18 | 0.75 |
| Gd (ppm) | 4.7 | 0.4 | 4.3 | 5.5 | 4.4 | 0.8 | 3.7 | 5.3 | 4.2 | 1.0 | 3.3 | 5.2 | 4.5 | 0.7 | 3.3 | 5.5 |
| Tb (ppm) | 0.9 | 0.1 | 0.7 | 1.0 | 0.7 | 0.2 | 0.6 | 0.9 | 0.9 | 0.4 | 0.6 | 1.3 | 0.8 | 0.2 | 0.6 | 1.3 |
| Dy (ppm) | 5.2 | 0.5 | 4.5 | 5.8 | 4.4 | 1.3 | 3.3 | 5.8 | 6.2 | 2.4 | 4.3 | 8.9 | 5.3 | 1.4 | 3.3 | 8.9 |
| Ho (ppm) | 1.1 | 0.1 | 0.9 | 1.2 | 0.9 | 0.3 | 0.7 | 1.2 | 1.3 | 0.5 | 0.9 | 1.8 | 1.1 | 0.3 | 0.7 | 1.8 |
| Er (ppm) | 3.3 | 0.4 | 2.7 | 3.8 | 2.7 | 0.7 | 2.1 | 3.5 | 4.1 | 1.6 | 2.8 | 5.8 | 3.3 | 0.9 | 2.1 | 5.8 |
| Tm (ppm) | 0.53 | 0.06 | 0.45 | 0.61 | 0.42 | 0.12 | 0.33 | 0.56 | 0.71 | 0.25 | 0.51 | 0.99 | 0.55 | 0.16 | 0.33 | 0.99 |
| Yb (ppm) | 3.6 | 0.3 | 3.1 | 4.1 | 3.0 | 0.9 | 2.4 | 4.0 | 5.5 | 2.0 | 3.8 | 7.7 | 3.9 | 1.4 | 2.4 | 7.7 |
| Lu (ppm) | 0.56 | 0.06 | 0.47 | 0.64 | 0.45 | 0.09 | 0.38 | 0.55 | 0.89 | 0.32 | 0.62 | 1.24 | 0.62 | 0.23 | 0.38 | 1.24 |

It is noteworthy to highlight that, by comparing with the mean values of the composition of continental crust, namely 5.6 ppm and 1.3 ppm for Th and U values [42], the LOP granites exhibit significantly higher Th and U values, given the mean values of 14.1 ppm and 13.9 ppm, respectively. Works performed by Teixeira et al. [43]), show that Th and U are mostly concentrated in zircon, monazite, allanite, and xenotime.

When comparing geochemical data, particularly focusing on Th and U content, with the interpolation maps of Th and U isotopes, a strong positive correlation is observed. There is an enrichment of all elements in the ESE border of LOP, aligning with a highly fractured area. Additionally, looking at Figure 4, a distinct blue spot is evident near the northernmost outcrop of the Barragem granite, suggesting a higher concentration of U in this region, aligning with the occurrence of the granite with the highest U content (Table 2). As stated earlier, only the northern outcrop of the BA granite was surveyed, which could account for why the blue spot is only depicted in this outcrop and not in the most representative outcrop of the BA, located in the central zone of the pluton.

To better understand the elemental distribution in LOP granites, predictive maps of eU/eTh, eU/K, and eTh/K ratios were made (Figure 5a–c).

The predictive maps for eU/eTh, eU/eK, and eTh/eK confirm the high concentration of U and Th radioelements in LOP, as well as highlight the heterogeneous distribution of radioelements in LOP granites.

Given that radioactive decay generates heat, the radioactive heat production rate was calculated for LOP. Considering the mean density of LOP as 2557 kg m$^{-3}$ [5] and using mean values of 9.1 ppm, 22.5 ppm and 4.5% for the U, Th, and K content, respectively, the heat production (*H*) of LOP is determined to be 4.09 µW m$^{-3}$.

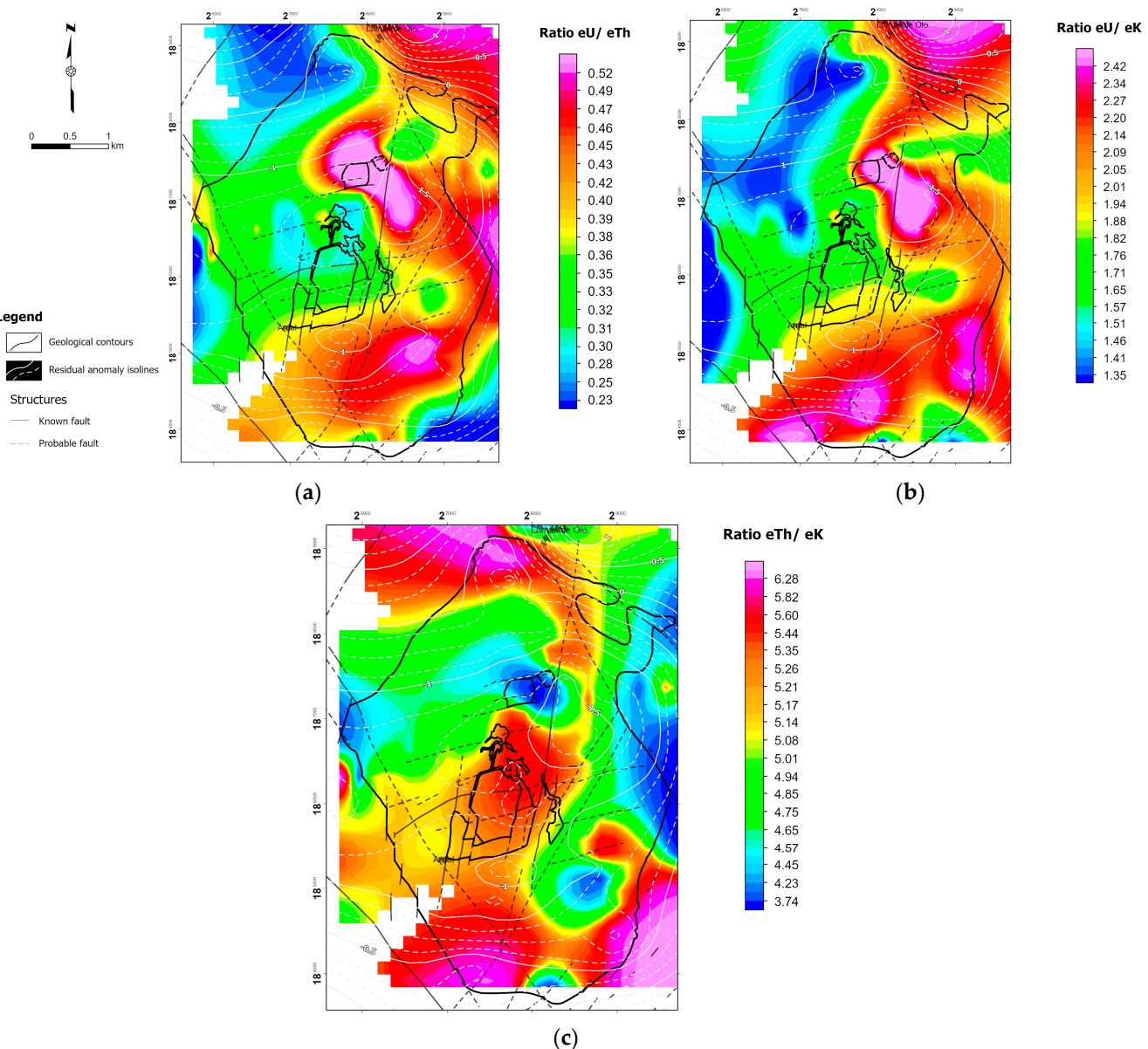

**Figure 5.** Predictions maps of (**a**) eU/eTh, (**b**) eU/eK, and (**c**) eTh/eK ratios (maps made in Oasis Montaj 2023.2 software).

Comparing the *H* of LOP to the mean value observed in granites worldwide, the LOP exhibits higher mean values. This suggests that the intrusion of this pluton has resulted in moderated heat flow in the crust; however, this is not enough to develop significant geothermal energy resources potential.

### 4.2. Magnetic Susceptibility

The magnetic susceptibility values show a heterogeneous behavior over the pluton (21 μSI < $K_m$ < 44,382 μSI), with both high (≥1000 μSI) and low (<1000 μSI) values of magnetic susceptibility ($K_m$). The Lamas de Olo granite shows a higher variability in the $K_m$ values, and the Barragem granite has the one that has the lowest values (e.g., [5,14,15]). The $K_m$ higher values (≥1000 μSI) are located at the east border of LOP, as well as in the central zone of the pluton (red and pink areas; Figure 6).

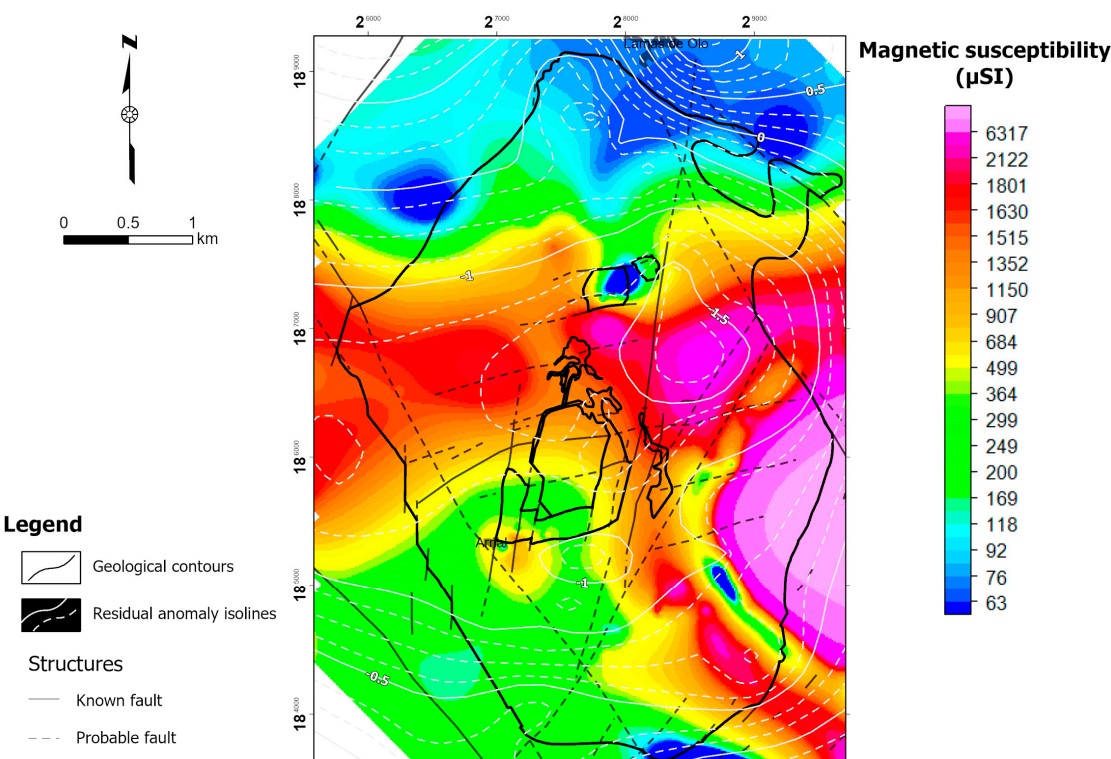

**Figure 6.** Magnetic susceptibly map of the Lamas de Olo Pluton (Magnetic susceptibility data interpolation made by minimum curvature in the Oasis Montaj 2023.2 software; modified from Cruz et al. [5,14,15]).

According to the classification of Ishihara [44], LOP belongs to the group of magnetite-type granites (e.g., [5,14,15]). The $K_m$ data, combined with petrographic, micro-Raman spectra, and other magnetic mineralogy studies (e.g., thermomagnetic curves and isothermal remanence magnetization), show the presence of pure magnetite, Ti-poor magnetite, and hematite in granites with high and low $K_m$ values, albeit in different proportions [14]. Beyond that, the petrographic studies show martitization processes which allow us to infer that LOP granites have originally a magnetite-type behavior, but later processes lead to a partial or total alteration of magnetite into hematite [5,14].

Comparing the spatial distribution of magnetic susceptibility (Figure 6) with the diverse radiometric maps (Figures 2–5), no straightforward correlation is evident. These data highlight, once again, the heterogeneous nature of LOP granites, characterized by a wide range of magnetic susceptibility values and a varied distribution of radioelements, as well as geochemical concentrations (Table 2).

*4.3. Gravimetric Survey*

The gravimetric survey was an efficient way to identify discontinuities, establish the geometry and depth of LOP granites, and infer about the pluton feeder zones. The Bouguer anomaly for each gravimetric station was calculated (using a reduction density of 2.66 g/cm$^3$), and the Bouguer anomaly map was obtained. The Bouguer anomaly in the studied area ranges between −37.53 mGal and −16.22 mGal, having −31.27 mGal as the mean value [5]. The regional and residual anomalies were calculated using 1st, 2nd, and 3rd order polynomial surfaces. Combining the residual maps with geological information (such as geological contacts, LOP outcrop shape, and existing faults), the model that best fits the surface of the Bouguer anomaly was identified as the residual anomaly map obtained through a 3rd order polynomial surface (Figure 7; [5]).

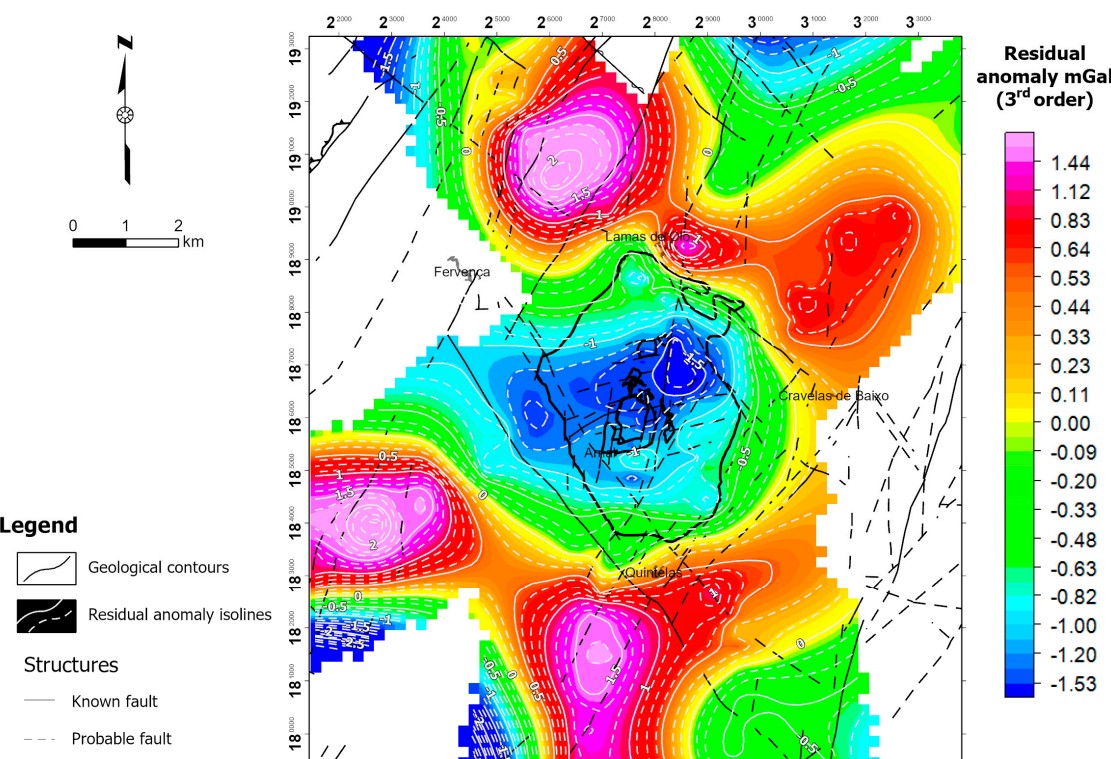

**Figure 7.** Residual anomaly map (mGal) of Lamas de Olo pluton and host rocks (residual map interpolated through the Oasis Montaj 2023.2 software and modified from Cruz et al. [5]). The white lines represent the isolines of the residual anomaly (3rd order) and are superimposed on the maps in Figures 2–6 to help the interpretation of the radiometric data.

The low negative anomalies are spatially correlated with the presence of less dense rocks, specifically granites (LOP granites and Vila Real two-mica granites). Conversely, the high values, predominantly in the southwest area, correspond to the presence of rocks with high-density values, such as those belonging to the Douro Group Metasedimentary rocks (e.g., schists; Figure 7).

The lower values in the central zone of the residual map are interpreted as the location of the LOP's feeding zone. Two deeper zones are identified: one in the NE zone of the LOP, and another one on the WSW border of the pluton, slightly displaced to the west of the current contours of the pluton (Figure 7). The location of the feeder zone was deduced based on the lowest residual anomaly values and local geological knowledge, specifically the intersection of late-Variscan fault systems (ENE-WSW with both NNE-SSW and NNW-SSE; [5]).

The correlation, between the radiometric values and the residual anomaly values, was carried out by XY-plots (Figure 8). These plots represent data collected at sampling sites where both radiometric and gravimetric values were acquired, accounting for 44 sites. Figure 8 provides a clear depiction that K isotopes exhibit the least variability across the study area, while Th and U values display significant variations throughout the pluton. However, no clear relationship between radioelements distribution and residual anomaly values was evident, emphasizing the heterogeneous distribution of radioelements across the pluton.

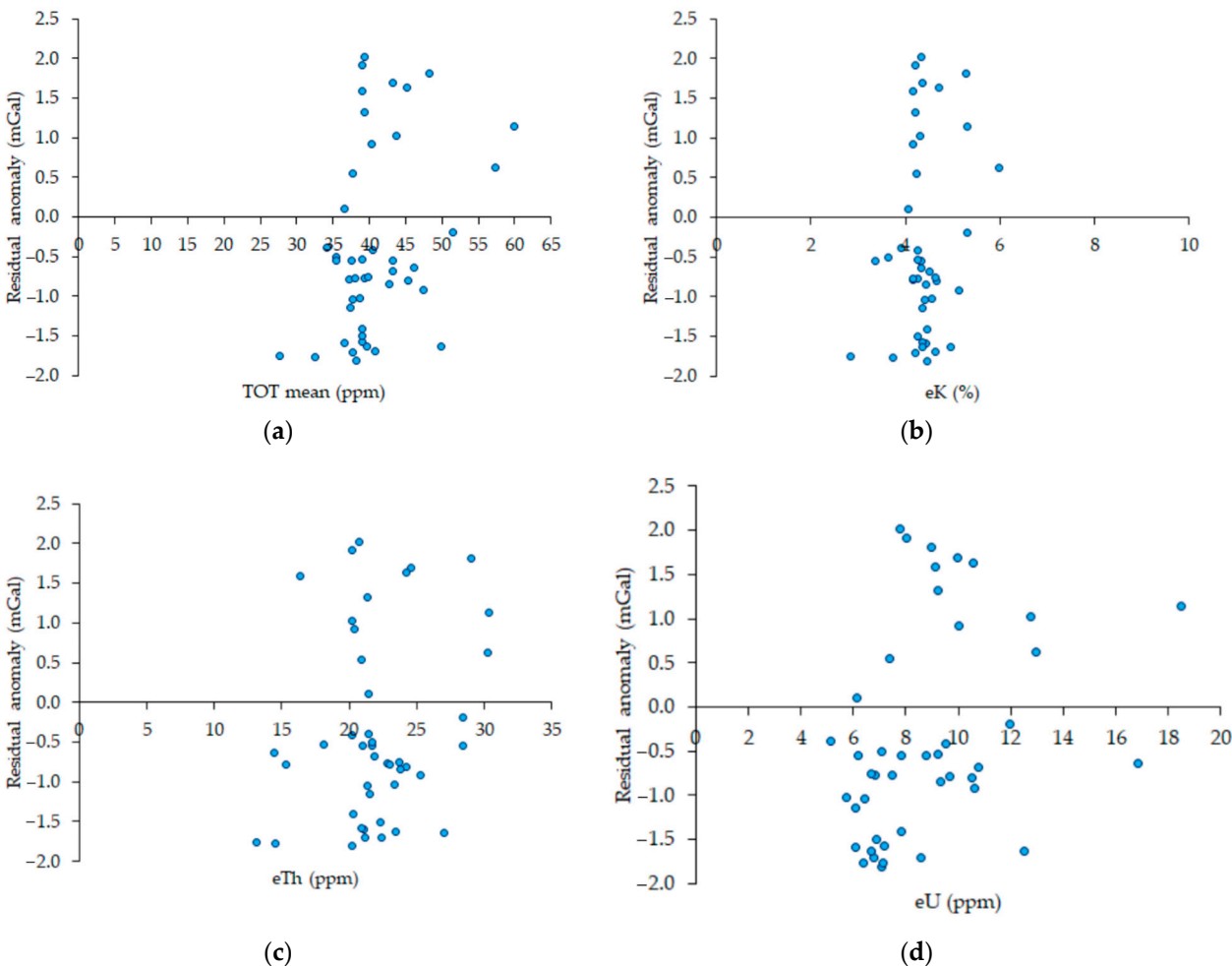

**Figure 8.** XY-plots of radiometric values versus (vs.) residual anomaly values (3rd order in mGal) of Lamas de Olo pluton and host rocks: (**a**) TOT mean (total radiation in ppm) vs. residual anomaly; (**b**) eK (values of K isotope in %) vs. residual anomaly; (**c**) eTh (values of Th isotope in ppm) vs. residual anomaly; (**d**) eU (values of U isotope in ppm) vs. residual anomaly.

To help our interpretations, the isolines of the residual anomaly (Figure 7) were superimposed on the various radiometric maps. Therefore, it can be concluded that:

- There is a positive spatial correlation between the northern feeder root (at the NE zone) and the higher concentration in the U isotope (Figures 4 and 5a,b);
- A positive spatial correlation between the western feeder root and the relative concentration of the Th radioelement, when comparing with K and U (Figures 4 and 5a,c).

Although no direct numerical relation between the K-Th-U and the residual anomaly values (Figure 8) was observed, there is a clear spatial relation between the higher concentration of U and Th in the feeder zones of the BA and LO + AC granites, respectively.

The distribution of U and Th isotopes in the feeder zones corresponds with the results outlined by Cruz et al. [5]. Their study indicates that the Barragem granite, characterized by a higher uranium content, primarily ascends through the NE root, using the intersection of NNE-SSW and WNW-ESE structures. Conversely, the ascent of Lamas de Olo and Alto dos Cabeços granites, richer in Th, predominantly occurred from the west zone root, guided by the intersection of W-E and NW-SE faults.

## 5. Conclusions

From this study, the following conclusions have been reached:

1. Radiometric measurements obtained through portable gamma-ray spectrometers effectively reveal the compositional heterogeneities within granitic plutons;
2. The LOP's radioactive heat production rate is 4.09 $\mu W\ m^{-3}$, which is higher than the global average for granites;
3. Feeder zones, identified in the residual anomaly map, can be correlated with the surface location of high-U and high-Th zones, corresponding to the northern and western roots, respectively. This correlation is in line with the geochemical character of the Barragem granite (rich in U), as well as the Lamas de Olo and Alto dos Cabeços granites (rich in Th);
4. Insights into granite petrogenesis and emplacement conditions can be gained through the examination of variations in isotopic distribution illustrated in ternary radiometric maps. These maps provided a "frozen image" capturing the final stages of magmatic construction.

**Author Contributions:** Conceptualization, C.C., H.S. and F.N.; methodology, C.C.; software, C.C.; validation, C.C., H.S. and F.N.; formal analysis, C.C., H.S. and F.N.; investigation, C.C., H.S. and F.N.; resources, C.C.; data curation, C.C.; writing—original draft preparation, C.C.; writing—review and editing, C.C., H.S. and F.N.; visualization, C.C., H.S. and F.N.; supervision, H.S. and F.N.; project administration, H.S.; funding acquisition, C.C. All authors have read and agreed to the published version of the manuscript.

**Funding:** The first author is a contracted researcher under the UIDP/04683/2020 project (DOI: 10.54499/UIDP/04683/2020) award by FCT—Foundation for Science and Technology, I.P. This work is supported by national funding awarded by FCT—Foundation for Science and Technology, I.P., projects UIDB/04683/2020 (DOI: 10.54499/UIDB/04683/2020) and UIDP/04683/2020 (DOI: 10.54499/UIDP/04683/2020).

**Data Availability Statement:** Data are contained within the article.

**Acknowledgments:** The authors thank the Unidade de Recursos Minerais e Geofísica from the Laboratório de Geologia e Minas—Laboratório Nacional de Energia e Geologia for providing radiometric measurements to the authors. The authors also thank Ricardo Ribeiro and António Oliveira for their help in the acquisition of radiometric data in the field. The authors express their gratitude to the anonymous reviewers for dedicating their valuable time, expertise, and providing constructive feedback that contributed significantly to the improvement of this article. The authors also thank Chuthathip Mangkonsu and Jingjing Yang for their guidance and insightful recommendations during the review process.

**Conflicts of Interest:** The authors declare no conflicts of interest.

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
