# Peer review of "A Multi-Method Approach to Geophysical Imaging of a Composite Pluton in North Portugal"

_minerals, doi:10.3390/min14040342_

Round 1

Reviewer 1 Report

Comments and Suggestions for Authors

Dear Editor

The paper submitted by Cláudia Cruz and coauthors introduces the application of direct radiometric measurements as well as some derived from previous regional radiometric studies to a composite pluton from NW Iberia. Both data sets fit well, despite the quality and reliability of the data provided in the paper is much more reliable and complete (K, U and Th are given separately). Besides, the authors also compared the radiometric data with previous data produce and interpreted by them; gravimetry and magnetic susceptibility. All in all, the paper provides a nice example of a multiple geophysical approach focused on understanding granite emplacement and mineralization processes. The paper is well-written and structured, methodology is correct and my only complaints deal with the processing of information. In any case, the paper merits publication after minor revisions.

Some comments/improvements:

1) Sampling spatial bias. Following figure 1 lets clear that the authors have carried out a sampling design in sections, mostly N-S and E-W. This is likely caused by outcropping and/or accessibility conditions, no problem with that., but this, and despite the interpolation algorithm used, undoubtedly condition the reliability of the mapping of variables. The authors should explain a bit this scope and be realistic with the implications along the study.

2) Numerical correlations are missed. Spatial comparisons of variables should be preferably focused on the sampling sections (but not aways from them, especially in the NW and SW sectors with little or no data). For this reason, I may suggest the authors to perform some XY-plots of the compared variables (e.g. eU vs Bouguer Residual anomaly). This exercise may help reinforcing and highlighting their inferences.

3) Fractures, joints and faults. The authors have done a great job in this granite; Cruz et al., 2021-JSG also provides a nice map on the fault and fracture networks. This map may be merged or joined in figure 1 to help the reader follow some arguments wielded along the paper

Minor changes

line #125. Specify the brand of the scintillator; Saphymo???

line #126. Specify, if possible, model and brand of the spectrometer

Figure 2: Specify units in the figure caption

lines #181-186. Please give here more details and add key references. I guess the concentration of the three elements must be somehow normalized since weight in % (K) and ppm (U and Th) cannot be straightly compared (maybe I am wrong).

Table 1: I wonder about the apparent no-significant difference between the granite and the host rock. I would also like to see the host rock data separately (although I believe they are very similar)

line #242. “clark values of the continental crust”, refers to Clark et al., 2011, if so adde the reference and correct the caption

lines #290-292. Surprisingly, the Bouguer and its residual are very little (20mGal and <4m mGal respectively) but the map has some sense. Please add mGal units in the color scale at figure 7.

lines #315-318: These sentences (as well as the conclusions) could be better supported by the mentioned XYplots

References:

Clark, C., Fitzsimons, I. C., Healy, D., & Harley, S. L. (2011). How does the continental crust get really hot?. Elements, 7(4), 235-240.

Cruz, C., Sant’Ovaia, H., Raposo, M. I. B., Lourenco, J. M., Almeida, F., & Noronha, F. (2021). Unraveling the emplacement history of a Portuguese post-tectonic Variscan pluton using magnetic fabrics and gravimetry. Journal of Structural Geolo

Reviewer 2 Report

Comments and Suggestions for Authors

The manuscript presents interesting results of the geophysical, particularly radiometric, investigation of a composite granitic intrusion in Northern Portugal. It demonstrates that the detailed geophysical study could add an additional dimension to understanding of the geology of granitic intrusions. I have two comments how to improve the manuscript.  (1) There should be more information on the geochemistry of the plutons so that the readers do not have to search literature to get basic information on the geochemistry. (2) The second comment regards heat flow production. The paper reports detailed information on radiometric data and thus should comment on the radiogenic heat production. What are the average radiogenic heat production values for the individual intrusions. Could the intrusions be classified as a “hot crust” regimes? Do the results suggest that the pluton has significant potential for geothermal energy resource development?

A procedure how to calculate heat production rate can be find in a number of papers. One of them is Rybach, L. Determination of heat production rate. In Proceedings of the Handbook of Terrestrial Heat Flow Density Determination; Hänel, R., Rybach, L.L., Stegena, Eds.; Kluwer: Dordrecht, The Netherlands, 1988; pp. 125–142.

Comments on the Quality of English Language

Moderate to minor English editing recommended
